# A Preprocessing Perspective for Quantum Machine Learning Classification Advantage in Finance Using NISQ Algorithms

**DOI:** 10.3390/e24111656

**Published:** 2022-11-15

**Authors:** Javier Mancilla, Christophe Pere

**Affiliations:** 1Stafford Computing LLC, 16192 Coastal Highway, Lewes, DE 19958, USA; 2INTRIQ, Department of Computer Science and Software Engineering, Université Laval Québec, Québec, QC G1V 0A6, Canada

**Keywords:** quantum machine learning, quantum data encoding, classical encoding, dimensionality reduction

## Abstract

Quantum Machine Learning (QML) has not yet demonstrated extensively and clearly its advantages compared to the classical machine learning approach. So far, there are only specific cases where some quantum-inspired techniques have achieved small incremental advantages, and a few experimental cases in hybrid quantum computing are promising, considering a mid-term future (not taking into account the achievements purely associated with optimization using quantum-classical algorithms). The current quantum computers are noisy and have few qubits to test, making it difficult to demonstrate the current and potential quantum advantage of QML methods. This study shows that we can achieve better classical encoding and performance of quantum classifiers by using Linear Discriminant Analysis (LDA) during the data preprocessing step. As a result, the Variational Quantum Algorithm (VQA) shows a gain of performance in balanced accuracy with the LDA technique and outperforms baseline classical classifiers.

## 1. Introduction

Machine Learning (ML) is a predominant tool nowadays to solve several challenges in different industries, such as credit scoring [1], fraud analysis [2], product recommendation [3], and demand forecasting [4], among other extensively explored use cases. Under this premise, the research of the quantum computing properties applied to ML has expanded rapidly in recent years since a proven advantage could be a highly useful cross-industry.

The recent progress of these explorations in Quantum Machine Learning (QML) [5] put a spotlight on quantum technology, introducing a challenge to determine if QML will provide an advantage over classical machine learning or not. The actual devices are noisy, meaning that the depth or consecutive gate operations are limited [6,7,8]. Qubits will lose their entanglement, and so will also lose the information. These devices make up the NISQ era [9] and limit the use of quantum algorithms or hybrid algorithms to be useful [10].

A few cases are already on the market, showing promising results and some companies’ commitment to the quantum machine learning journey. One example is CaixaBank (Spanish Bank), which is working and testing QML models using the Pennylane quantum framework to define a scoring model for risk assessment [11].

One of the major concerns to even grasp a reliable result remains on the input/output concept and on the number of available good qubits to use. The bound of 100+ was reached by IBM [12], but it is still insufficient to use complex algorithms that require thousands or millions of qubits depending on the type of problem to be addressed [13].

To be practical in a business context, QML techniques need to avoid the small-number-of-qubits constraint and create a methodology to use big datasets on the current NISQ devices. Previous works showed great potential in splitting big circuits and learning the weights of the different gates separately or reusable qubits for image classification [14].

In this paper, we approach the input problem by comparing different preprocessing and classifier methods on small and larger datasets with a binary target. The objective is to determine a specific architecture for preprocessing, dimensionality reduction of the dataset structure, the encoding manner and the corresponding classifier. We demonstrate that using Linear Discriminant Analysis (LDA) within the preprocessing phase is better than Principal Component Analysis (PCA) when the dataset possesses an important number of features. We generalize this approach by studying the effect of LDA on the encoding qubits.

Different tabular datasets (Section 2) are used to understand the link between the number of features and the encoding process (Section 5). We develop a pipeline (Section 3) to compare the classical and quantum classifier. This study leads us with a review of the current literature to determine and discuss rules to obtain a quantum advantage with the current NISQ devices.

## 2. Datasets

The dataset selection in this research aims to emulate real business case scenarios where the users can find imbalanced dataframes, a small or large number of features and also represent—in this case—the financial behavior of a group of people. We used well-known datasets extracted from UCI and Kaggle to achieve more than 100 features per datapoint in a CSV file for one of the cases. Nevertheless, we wanted to use an even larger dataset, but it was extremely difficult to find information with more variables and features in a public and open license manner.

### 2.1. UCI—Default of Credit Card Clients Dataset

The dataset (https://www.kaggle.com/datasets/uciml/default-of-credit-card-clients-dataset (accessed on 3 July 2022)) contains information about credit card clients collected in Taiwan from April 2005 to September 2005 [15]. The data possesses 25 features and 30,000 rows corresponding to individual clients. This data was extracted from Kaggle but is originally stored in the well-known University of California, Irvine (UCI) dataset repository.

The objective of using this dataset is to deal with a classification problem assessing the prediction of default under credit card usage. This data is imbalanced, having 22% of defaulters and 78% of non-defaulters. The variables have demographics, payments, billing and current credit card information features.

In this research, the dataset was used as it is and without modifications such as oversampling, undersampling, SMOTE or any previous transformation until we applied the preprocessing designed for the quantum pipeline.

### 2.2. Fraud Detection Dataset

This dataset (https://www.kaggle.com/datasets/volodymyrgavrysh/fraud-detection-bank-dataset-20k-records-binary (accessed on 3 July 2022)) reflects the information of bank fraud transactions on 20,468 datapoints and 114 features. This data was extracted from Kaggle and was originally uploaded by Volodymyr Gavrish (https://www.kaggle.com/volodymyrgavrysh (accessed on 3 July 2022)).

The objective of using this dataset is to identify which users are fraudsters or not due to the transactional information. The data is imbalanced with 27% under class 1 and 73% in class 0.

In this research, the dataset was used as it is as well and without modifications until we applied the preprocessing designed for the quantum pipeline.

An important point to be highlighted in this dataset is that it looks very similar to a real-world scenario (+100 features and 1000s of data points), but we didn’t manage to confirm the source of the information; so, the assumption is that it must be synthetic or generated. Anyway, this file has more than 2000 downloads and 19,000 views on Kaggle and remains one of the most-used dataframes to explore classification techniques.

## 3. Methods

### 3.1. Dimensionality Reduction

PCA is one of the predominant structures for dimensionality reduction in the exploration of classic data into QML algorithms [16]. This technique is used to reduce the features and compress them into *N* variables to match a set of *N* qubits available to run a classification algorithm using a gate-based quantum circuit. This method is commonly used for unsupervised linear transformation and to find the maximum variance in high-dimensional data. PCA reduces dimensions by examining the correlation between various features, creating orthogonal axes, or principal components, with the direction of maximum variance as a new subspace.

There are many alternatives to PCA, but one of them demonstrates a significant impact when we are dealing with quantum classification problems. LDA is a supervised method that considers class labels by reducing the number of dimensions. LDA seeks to identify a subspace of features that optimizes class separability optimally. LDA operates by computing the d-dimensional mean vector for each class label and then constructing a scatter matrix within each class and between them.

As we mentioned, both PCA and LDA are linear transformation techniques that decompose matrices into eigenvalues and eigenvectors. PCA is unsupervised and does not consider class labels, whereas LDA does. These two techniques will be applied to classical data preprocessing for QML. We will demonstrate the advantage of the LDA surpassing PCA under a small number of qubits and with relevant features.

### 3.2. Training

The interest in QML is located in the quantum advantage that can be brought by such new technology. A quantum advantage appears when the quantum algorithm approach provides better or faster results than the classical equivalent. In this paper we benchmark classical classifiers such as Logistic Regression [14], Decision Tree [17], Random Forest [17], K-Nearest Neighbors [17], SVM [17], Quantum Kernel (QSVC) [18], and the Variational Quantum Algorithm [19]. The models are benchmarked with k-fold cross-validation with 10 folds. Table 1 shows the metrics used to evaluate these algorithms.

In a classification task, TP stands for True Positive when the model predicts the right positive value. TN stands for True Negative when a model correctly predicts a false value. False positive is when the model predicts true instead of false, and FN is for false negative when the model predicts false instead of true. With these values, it is possible to use precision, the metric that quantifies the number of positive predictions made. The recall is the metric to quantify the number of positive predictions made through all the positive predictions that could be made. The f1-score combines precision and recall to capture both pieces of information. The Matthews correlation coefficient of the Phi coefficient is a metric which uses all four values of the confusion matrix to evaluate the behavior of a classifier; this metric is stronger than the previous ones. Balanced accuracy is the mean of sensitivity and specificity, it is used to evaluate how strong a classifier is.

## 4. Backends

Quantum computers and simulator backends are not trivial decisions when a fast iteration is needed with larger datasets. Typically, machine learning models need several adjustments and iterations until we put them in production or under real-world operation (fine-tuning). In the case of QML, the challenge is the same, but the hardware ecosystem is different. A quantum algorithm can be run on a simulator (simulation of a perfect and noisy quantum computer) and real devices.

### Simulators

Mainly the use of quantum simulators allows us to test and evaluate the results under potential real quantum computation scenarios and typically gives us a chance to operate up to 50 qubits using classical computers. In the case of this experiment, we are using the Qiskit Aer simulator and the default qubit simulator device from Pennylane [20] only.

Qiskit Aer is a high-performance simulator for Qiskit Terra that provides a highly adjustable noisy model for investigating quantum computing in the NISQ domain. The core is designed in C++ for speed and includes elements from IBM’s high-performance online simulators into a local simulator that is scaled to operate even on your own laptop or server.

Pennylane’s default qubit simulator is a simple state-vector designed in Python (with JAX, Autograd, Torch, and Tensorflow). This simulator is recommended by Pennylane for optimization with a reduced number of qubits or when stochastic expectation values are going to be used.

The objective behind using simulators alone in this research is that we can have fast iterations, adjustments, and results. Our choice does not mean that our models’ code and structure cannot be applied to real quantum hardware. The main factors that make the difference are the time (speed) and the noise since simulators run in a fraction of the time compared with real quantum computers (in the case of the QML architectures used in this investigation) and also without noise depending on the configuration. In our case, we subtracted the noise from the simulations.

## 5. Algorithms

### 5.1. Machine Learning Models

Classification problems are part of the supervised learning domain, and that is why we used several classical algorithms in this subarea of ML to set a benchmark against the hybrid quantum-classical approach.

#### 5.1.1. Logistic Regression

This method is one of the simplest for the binary classification problem [21]. The model is trained to learn the parameters of the linear equation ${\hat{y}}^{\left(i\right)}=\beta_{0}+\beta_{1}x_{1}^{\left(i\right)}+\beta_{2}x_{2}^{\left(i\right)}+\ldots +\beta_{n}x_{n}^{\left(i\right)}$ where $\beta_{n}$ are the coefficients of the linear regression, $x_{n}$ are the features for the *i* sample. Linear regression is used in regression tasks, but the method can be applied in the classification task by a subtle embedding in a logistic function such as Equation (Equation 1) to compute a probability.
(1)$$P\left(y^{\left(i\right)}=1\right)=\frac{1}{1+e^{-(\beta_{0}+\beta_{1}x_{1}^{\left(i\right)}+\ldots +\beta_{n}x_{n}^{\left(i\right)})}},$$
where *P* is the probability that the label *y* for the sample *i* corresponds to the value 1. The probability is computed for each sample (the model learns the corresponding coefficients) and the probability threshold is fixed at 0.5 to separate the binary outcome. If $P(y^{\left(i\right)}=1)<0.5$, the corresponding label is 0; if $P(y^{\left(i\right)}=1)\geq 0.5$, the label is 1. The logistic regression method requires a lot of samples to be stable to efficiently approximate the $\beta_{n}$ parameters.

#### 5.1.2. Decision Tree

A decision tree also called “Classification and Regression Trees” (CART), is a sort of binary graph where the next child is based on the previous decision. The base of the tree is the root, and then two branches are created, which are split into categories of “yes” or “no”. A tree structure is built by successive decisions until the latest, called the leaf, is reached. This type of technique is simple but prone to overfitting. They are powerful algorithms capable of fitting complex datasets. The learning process is done with the Gini criteria or entropy (Equation (Equation 2)).
(2)$$H_{i}=\sum_{k=1}^{n}P_{i,k}\log_{2}P_{i,k},$$
where *i* is the *i*th node, $P_{i,k}$ is the probability of the category *k*.

#### 5.1.3. Naïve Bayes

The Naïve Bayes or NB algorithm is a simpler version of the Bayes theorem (Equation (Equation 3)).
(3)$$P\left(AB\right)=\frac{P(BA)\cdot P(A)}{P(B)},$$
where *A* and *B* are events, P(AB) is the probability of *A* given *B* is true, P(BA) is the probability of *B* given *A* is true, P(A) and P(B) are the independent probabilities of A and B, respectively. In the case of the NB classifier, the probabilities are conditionally independent. It significantly reduced the computation and transformed it into a tractable problem.

#### 5.1.4. k-Nearest Neighbors

The k-Nearest Neighbors (k-NN) is a simple non-parametric distance-based algorithm. The hypothesis is that a similar point will be closed in an n-dimensional space. A point will be encoded and positioned by distance computation (e.g., Euclidean distance). Then, the algorithm takes the *k* nearest neighbours and computes the classes’ average to predict the corresponding class for that new point.

#### 5.1.5. SVM

Support Vector Machines (SVM) are a class of algorithms based on class separation by a plan. An SVM will create a plan to create a binary separation between the classes. Then, the algorithm will compute the distance of each point and plan to maximize the distance. When the classes are not linearly separable, SVM can be used with kernels. Kernels are a trick to compute small successive plans to separate classes in complex datasets. They are particularly efficient in high-dimensional space.

### 5.2. Dimensionality Reduction

#### 5.2.1. SVD

Singular Value Decomposition, or SVD, was established to decompose the matrix representation of data into distinct matrices. It’s a factorization process for real and complex matrices. These transformations are based on eigenvalue decomposition to diagonalize a matrix.

#### 5.2.2. PCA

Principal component analysis or PCA is a widely used method for dimensionality reduction in the context of machine learning. The objective is to transform a large dataset (a high number of features) into a compact representation containing the data’s important information (orthogonal projection). Reducing the dimension is closely related to loss in accuracy, but PCA as SVD uses the eigenvalue decomposition process to transform the covariance matrix involved in the process. The components are a linear combination of various features to create uncorrelated new features. The first component will contain the maximum amount of the information; then, the remaining information will be contained in the second, etc. The geometrical representation of PCA is that components represent the direction of the maximal amount of variance (rotation).

#### 5.2.3. SKPP

Projection pursuit is a generalization of PCA [22] where the method aims to find the best projections through the feature to maximize or minimize a projection index. To find the relevant project index, the method uses the Kurtosis-based projection [23]. In a case of a supervised dataset, the algorithm is named Supervised Kurtosis Projection Pursuit (SKPP). The Kurtosis index can be expressed as:(4)$$K=\frac{1/n\sum_{i=1}^{n}{\left(z_{i}-\tilde{z}\right)}^{4}}{{\left(1/n\sum_{i=1}^{n}{\left(z_{i}-\tilde{z}\right)}^{2}\right)}^{2}},$$
where *n* is the number of samples, $z_{i}$ is the individual sample value, and $\tilde{z}$ is the sample mean. The numerator is the fourth central moment, and the denominator is the biased sample variance.

#### 5.2.4. LDA

Linear Discriminant Analysis is similar to PCA; they are linear transformations to reduce the dimensionality of datasets (eigenvalue decomposition). Where PCA will maximize the variance, LDA will maximize the axes for class separation. LDA will create a subspace of k-dimensions from the n-dimensions space of the original data where k≤n−1. The subspace is computed taking into account the label to maximize the separation of classes.

### 5.3. Quantum Machine Learning Models

Quantum Computing (QC) and Machine Learning (ML) have been mixed to develop the new area of Quantum Machine Learning (QML). This new field of study incorporates ideas from both aspects to provide better answers by boosting the performance of either ML algorithms or quantum experiments, or both. By utilizing quantum resources to increase machine learning in terms of speed and/or performance, researchers could obtain potential alternative and/or more accurate solutions.

#### 5.3.1. Quantum Kernel

The quantum kernel methods [24,25] in principle are the same as their classical versions, which aim to classify data by defining what is similar in a given space because of their distance using a feature mapping function. The main difference in the quantum version of kernels is that it maps out the data points from the original input to a high-dimensional Hilbert feature space, expanding the possibilities to find the best classification possible [26]. One of the well-known classical methods that utilize the kernel properties is the support vector machine (SVM), also known as support vector classifier (SVC), which is dedicated to finding a hyper-plane that can separate the classes of the datapoints, expanding as much as is possible the distance between both groups.

In this research, we use a similar structure of an SVC, but ours is boosted with a quantum kernel. The mechanism of quantum kernel functions resembles the conventional one, but its implementation relies on quantum superposition states and entanglement. Also, in the case of quantum kernels, the output values are statistically dependent on probabilities, so some researchers call this method a probabilistic kernel function.

#### 5.3.2. Variational Quantum Classifier

A Variational Quantum Classifier (VQC) [27,28,29] is a supervised quantum-classical hybrid method widely used in NISQ devices and simulators. The cost function in the case of this algorithm is calculated using iterative measurements, which also provide the possibility of mitigating errors. This method allows the researcher and developers to map classical data and grab benefits for an increasingly ample feature space in quantum. The quantum execution for supervised learning employs variational algorithms that are implemented using differential programming, state preparation that encodes classical data sets into amplitude and rotations of qubits for quantum hardware or simulators to comprehend, and qubits that are executed using parameterized unitary operations; all parameters are modifiable according to given rules. The outcome of the quantum execution is the output, which categorizes the input data.

### 5.4. Quantum Encoding

Quantum encoding is the process of passing from classical data to quantum representations. There are many ways to process classical data and create a useful representation. In this study, we used a quantum feature map (Qiskit ZZFeatureMap) and angle encoding (Pennylane) to be used with QSVC and VQC, respectively.

#### 5.4.1. Quantum Feature Map

A feature map is a new representation for data encoding [26,30,31,32] and it is represented as a Hilbert space where the transformation x→|ϕ(x)〉 is applied to pass from the original representation *x* into a linear separable space via a unitary operator ϕ(x). In other words, we encode the classical data into quantum states and map them to Hilbert space.

The feature map is a good (sometimes infinite) projection for using SVM that is designed to create a hyperplane between classes. This hyperplane is a linear separation of two subspaces that are automatically created by the feature map (different data projections make the separation easier with higher dimensions).

#### 5.4.2. Angle Encoding

Angle encoding is a process of encoding classical information by rotations [33]. The classical information is represented by angles of rotation in corresponding gates and can be written as Equation (Equation 5):(5)$$|x\rangle =\overset{n}{\underset{i}{⨂}}R\left(x_{i}\right)|0^{n}\rangle ,$$
where *R* is rotation gates such as $R_{x}$, $R_{y}$ and $R_{z}$. Angle encoding is used when the dimension of the feature vector *x* is equal to the number of qubits.

### 5.5. Workflow

In Figure 1, we present the workflow we used throughout this study to compare the selected algorithms (classical and quantum). The set of algorithms was applied to the data representation generated by dimensionality reduction methods. The workflow is composed of five steps:1–2Steps 1 and 2 can be associated: Load the data and apply an Exploratory Data Analysis. The objective is to clean the data and normalize it with a good format for the dimensionality reduction method.3Dimensionality reduction: SVD, PCA, SKPP and LDA are used to reduce the number of features to two compressed dimensions. SVD, PCA and SKPP were used with two components. LDA was used on a split dataset. Each half part was reduced with one component by LDA.4Quantum encoding: The classical data is encoded into a quantum representation by quantum feature maps. This step was only used for quantum algorithms.5Applied models: The selected set of algorithms (ML and QML) is applied to the data encoding (classical or quantum) and evaluated through the same metrics (Table 1).

During the workflow, we evaluate the set of selected algorithms based on the same sample of data. 800 data samples were used for the training process, and 200 were used for the test. Only two qubits were chosen through this study to show the usability of a small number of qubits in a business context. Two datasets close to the real world were selected to estimate the importance of current quantum algorithms with NISQ devices.

## 6. Results

This section will present the results obtained by applying the workflow to the two datasets we selected. The core of the analysis is to take the position of the business context. The classical machine learning algorithms are applied to both datasets, without dimensionality reduction, and serve as a baseline. Then, the quantum algorithms are applied to a subset of each dataset with diverse dimensionality reduction techniques. This choice is motivated by focusing on only two qubits to compare the results. Indeed, current commercial solutions provide quantum computers with two qubits. Also, cloud-free available quantum computers are up to five qubits.

We focus on a small number of qubits to demonstrate the usability of quantum algorithms in a business context. The quantum version of SVM (QSVC) and a variational quantum algorithm (VQA) have been used to challenge the classical machine learning methods. The subsample is constituted of 800 samples for the training phase and 200 for the test phase. Each model, classical and quantum, will be evaluated with the metrics presented in Table 1. Each metric for each algorithm is associated with an error bar determined by a k-fold cross-validation approach with 10 folds. Only the VQA was computed differently due to its implementation, and it is not provided with an associative error.

We will analyze the results for both datasets separately in the following subsections. Classical machine learning models were also applied to the same sample with the same dimensionality reduction approaches; the results are provided for comparison in Appendix A. These results are discussed in Section 7 Discussion.

### 6.1. UCI Credit Card Default Dataset

Table 2 shows the results for the baseline determined with the classical machine learning models. LR and SVM show a non-convergence state with predictions only for the majority class. Naïve Bayes classifier shows the smallest precision compare to CART and KNN. The baseline demonstrates that classical methods struggle to create a good classifier to separate the minority class from the majority class. The precision is at a maximum of 38.74% for the KNN, and the best f1-score is reached by CART with a small value of 39.10%. These poor metrics are the results of an extremely imbalanced dataset. Little information is provided by the minority class, which tends to complexify the classification process. Machine Learning methods are best suited for a balanced dataset, but this condition is rarely present in the industry.

Table 3 shows the results for QSVC and VQA applied to the UCI Credit Card dataset. They are used with SVD, PCA, SKPP and LDA dimensionality reduction techniques. QSVC with the SKPP technique shows metrics with 0.00%, meaning a non-separation between both classes. QSVC, in this case, predicts only the majority class output, ignoring the minority class. The other algorithms associated with the different dimensionality reduction approaches show a convergence. VQA with SVD, VQA with PCA, and VQA with SKPP provide interesting results, comparable to or better than the baseline. LDA is the best dimensionality reduction for both quantum algorithms. Both algorithms show the best results in each metric on the UCI dataset.

Figure 2 shows the difference between LDA and PCA methods for the UCI Credit Card dataset. The LDA representation provides a net advantage for quantum encoding. The use of LDA shows a quantum advantage for QSVC and VQA algorithms. Figure 3 shows a histogram representation of the metrics for both the ML and QML approaches.

### 6.2. Fraud (Bank) Detection Dataset

Table 4 shows the results for the baseline for the fraud (bank) detection dataset with classical ML. The precision is up to 70% for LR, KNN, and CART, but NB has a precision of less than 30%, and SVM shows metrics with 0.00%. Table 5 show the results for the quantum algorithms with the different dimensionality reduction techniques. PCA demonstrates the worst representation for QSVC (Figure 4). The algorithms did not converge. LDA shows the best results for both QSVC and VQA. SKPP also demonstrates interesting results for both algorithms. It is worth noting that four quantum algorithms beat the best precision value of the baseline and the other metrics are close to the classical ML models.

Even if the baseline classical ML models perform well, the quantum algorithms provide a small advantage over the LDA approach (as the UCI Credit Card fraud dataset results). Figure 5 shows VQA and QSVC compared to CART and KNN (metrics representation).

## 7. Discussion and Conclusions

In this study, we demonstrate that specific preprocessing techniques could play a crucial role when discussing quantum machine learning. We focus on the encoding part, the classical one, to evaluate the effect on quantum algorithms. We show that a quantum computer can extract more meaningful information from classical data and leverage classification results just using a few dimensions. As postulated in [34], quantum advantage does not need to be measured by the ability to beat classical ML models but can be regarded as a better information extraction technique. The few numbers of qubits of currently accessible quantum computers force researchers to look for new alternatives. Classical dimensionality reduction programs, such as SKPP, PCA or LDA, are useful to compress classical high feature datasets (100+) into a number that can be used with a quantum computer. Here, we tested with two dimensions for two qubits. LDA shows more promising results for supervised machine learning tasks with quantum computers. The prevalence of LDA under PCA was not explored in this paper but will be explored in the future to understand how LDA provides a better data representation for qubit encoding. Further analysis will be needed to determine the positive effect of LDA in supervised QML. Also, we will study the potential impact of PCA on unsupervised tasks. As in classical ML, we need to determine which methods have a better effect on specific types of data.

Table A1, Table A2, Table A3, Table A4, Table A5, Table A6, Table A7 and Table A8 show the results where the classical methods were also applied with the dimensionality reduction methods. The DR also improves the performances of these methods, but quantum algorithms are comparable to them. More investigation will be needed, but the preprocessing part of big real-world datasets plays an important role in the usability of quantum computing in the industry. Better quantum data encoding will also be required to demonstrate a strong difference between classical and quantum machine learning.

Classical ML methods selected for this study were not tuned specifically on the two datasets. Default parameters were chosen. Only LR was used with a max iteration parameter fixed at 1000 iterations. Indeed, on the fraud detection dataset, the default limit was reached, and LR was not converging. KNN was trained with the number of neighbours fixed at seven. In the case of quantum algorithms, the data is encoded with a feature map. The parameters of the feature map were not tuned but fixed through the study with a number of repetitions of 2 and a feature dimension of 2. Further investigation is needed to explore the effect of the feature map on the output of the QSVC. VQA was used with an angle embedding method (rotations gates) and a strongly entangled layer. Neither were tuned, and alternatives will be explored in the future. An interesting perspective will also be to study the impact of different ansatzes [35,36] on the VQA results.

Also, in this study, we used only two datasets close to the real-world data in the finance domain. The results demonstrate a quantum advantage with QSVC and VQA, but we need to extend the approach with other datasets (higher number of features) in other domains to create a benchmark through a general application. We demonstrate that the quantum era needs to be seriously investigated by the industrial people, but more work is needed to fully demonstrate the advantages. 

## Figures and Tables

**Figure 1 entropy-24-01656-f001:**
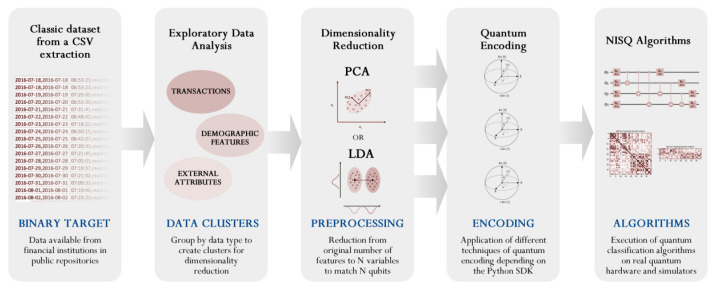
High-level workflow of the process for the hybrid quantum approach followed in this research. The input is a dataset in CSV format that is analyzed and divided depending on the following decomposition strategies to be applied. The next step is dimensionality reduction using different techniques, but mainly PCA and LDA as the main comparison in this exploration. Once the reduction is executed—to match with the qubits to be used in the algorithm—a quantum encoding must be conducted, and the following quantum algorithm must be applied to extract the results.

**Figure 2 entropy-24-01656-f002:**
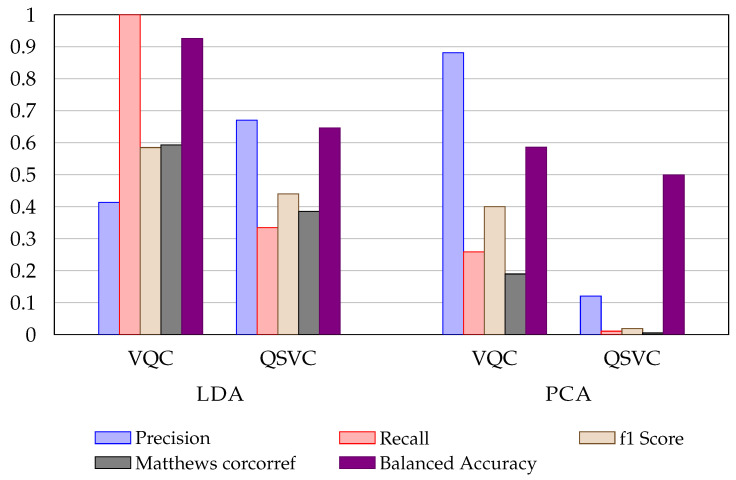
Metrics comparison between VQC and QSVC using LDA and PCA applied on UCI Credit Card default dataset.

**Figure 3 entropy-24-01656-f003:**
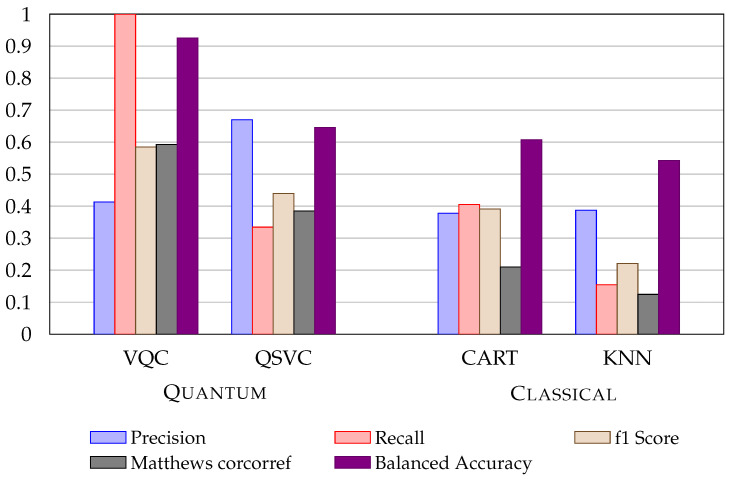
Metrics comparison of VQC and QSVC with CART and KNN (best classical algorithms) with the application of LDA using UCI Credit Cards dataset.

**Figure 4 entropy-24-01656-f004:**
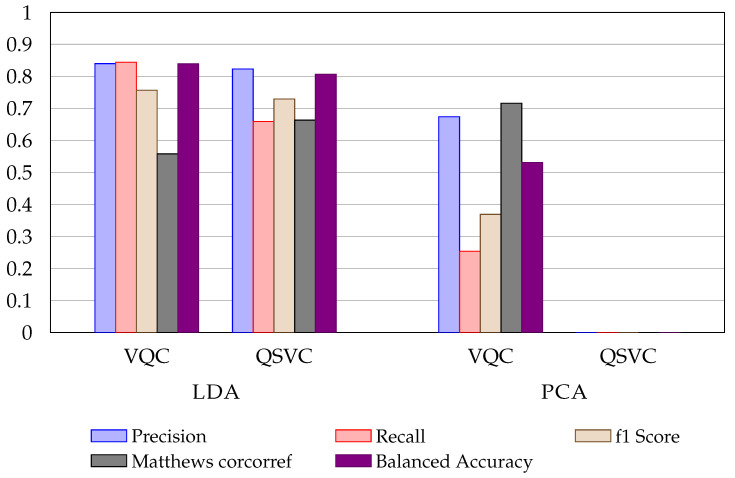
Metrics comparison between VQC and QSVC using LDA and PCA applied on fraud (bank) detection dataset.

**Figure 5 entropy-24-01656-f005:**
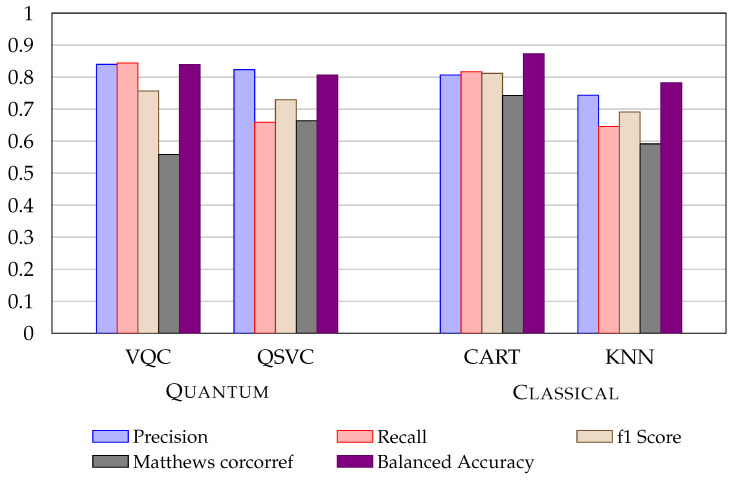
Metrics comparison of VQC and QSVC with CART and KNN (best classical algorithms) with the application of LDA using fraud (bank) detection dataset.

**Table 1 entropy-24-01656-t001:** Metrics are used to evaluate the different classifiers and their corresponding equation. Where TP, FP, TN, and FN are True Positive, False Positive, True Negative, and False Positive, respectively. $P_{o}$ is the probability of Observed agreement, and $P_{c}$ is the probability of chance agreement. MCC is the Matthews correlation coefficient.

Metric	Equation
Precision	$\frac{TP}{TP+FP}$
Recall	$\frac{TP}{TP+FN}$
F1-score	$\frac{2\times (Precision\times Recall)}{Precision+Recall}$
MCC	$\frac{TP\times TN–FP\times FN}{\sqrt{\left(TP+FP)(TP+FN)(TN+FP)(TN+FN\right)}}$
BA	$\left(\frac{TP}{\left(TP+FN\right)}/\frac{TN}{\left(TN+FP\right)}\right)/2$

**Table 2 entropy-24-01656-t002:** Baseline ML models for the UCI Credit Card default dataset. The results for logistic regression (LR), k-Nearest Neighbors (KNN), Naïve Bayes (NB) and Support Vector Machine (SVM) are computed with k-fold cross-validation with 10 folds.

	Precision (%)	Recall (%)	f1-Score (%)	Matthews	Balanced
				Corcorref (%)	Accuracy (%)
LR	0.00 (0.00)	0.00 (0.00)	0.00 (0.00)	−0.22 (0.44)	49.99 (0.01)
KNN	**38.74** (2.03)	15.45 (1.51)	22.07 (1.76)	12.43 (0.76)	54.26 (0.65)
CART	37.79 (1.51)	40.53 (1.51)	**39.10** (1.34)	**20.99** (1.45)	**60.76** (0.75)
NB	24.71 (0.89)	**88.41** (1.55)	38.62 (1.15)	11.94 (1.74)	55.82 (0.88)
SVM	0.00 (0.00)	0.00 (0.00)	0.00 (0.00)	0.00 (0.00)	50.00 (0.00)

**Table 3 entropy-24-01656-t003:** Quantum models applied on the UCI Credit Card default dataset with the corresponding dimensionality methods.

	Precision (%)	Recall (%)	f1-Score (%)	Matthews	Balanced
				Corcorref (%)	Accuracy (%)
QSVC (SVD)	20.00 (40.00)	2.21 (4.82)	3.92 (8.45)	5.98 (12.30)	51.10 (2.41)
VQA (SVD)	77.50	26.72	39.74	19.75	58.00
QSVC (PCA)	12.00 (29.93)	1.06 (2.14)	1.88 (3.84)	0.51 (8.04)	49.93 (1.30)
VQA (PCA)	**88.10 **	25.87	40.00	18.95	58.55
QSVC (SKPP)	0.0 (0.0)	0.0 (0.0)	0.0 (0.0)	0.0 (0.0)	50.0 (0.0)
VQA (SKPP)	25.58	27.5	26.51	7.3	53.75
QSVC (LDA)	67.02 (13.31)	33.44 (10.08)	43.96 (10.97)	38.51 (10.97)	64.6 (5.08)
VQA (LDA)	41.30	**100.00**	**58.46**	**59.28**	**92.54**

**Table 4 entropy-24-01656-t004:** Baseline ML models for the fraud (bank) detection dataset. The results for logistic regression (LR), k-Nearest Neighbors (KNN), Naïve Bayes (NB) and Support Vector Machine (SVM) are computed with k-fold cross-validation for 10 folds.

	Precision (%)	Recall (%)	f1-Score (%)	Matthews	Balanced
				Corcorref (%)	Accuracy (%)
LR	71.54 (2.77)	47.27 (1.96)	56.88 (1.62)	46.89 (1.88)	70.2 (0.88)
KNN	74.34 (1.77)	64.56 (2.36)	69.09 (1.91)	59.16 (2.65)	78.22 (1.39)
CART	**80.68** (1.87)	81.69 (2.06)	**81.17** (1.63)	**74.27** (2.25)	**87.28** (1.21)
NB	28.43 (1.07)	**96.95** (0.88)	43.96 (1.3)	12.58 (1.36)	54.07 (0.54)
SVM	0.0 (0.0)	0.0 (0.0)	0.0 (0.0)	0.0 (0.0)	50.0 (0.0)

**Table 5 entropy-24-01656-t005:** Quantum models applied on the fraud (bank) detection dataset in combination with the corresponding dimensionality methods.

	Precision (%)	Recall (%)	f1-Score (%)	Matthews	Balanced
				Corcorref (%)	Accuracy (%)
QSVC (SVD)	85.02 (11.42)	39.24 (8.53)	52.94 (8.19)	49.55 (7.54)	68.45 (3.97)
VQA (SVD)	62.50	72.22	60.61	26.57	74.09
QSVC (PCA)	0.0 (0.0)	0.0 (0.0)	0.0 (0.0)	0.0 (0.0)	50.0 (0.0)
VQA (PCA)	67.39	25.41	36.90	7.16	53.09
QSVC (SKPP)	56.28 (11.17)	46.46 (7.21)	50.3 (6.8)	35.53 (8.7)	66.65 (4.02)
VQA (SKPP)	**89.86**	68.89	**77.99**	**70.67**	82.60
QSVC (LDA)	82.35 (10.29)	65.92 (8.79)	72.93 (8.14)	66.35 (9.9)	80.67 (4.94)
VQA (LDA)	84.00	**84.44**	75.68	55.81	**83.92**

## Data Availability

The notebooks used during this study can be provided after contacting the authors.

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
