# Peer review of "A Preprocessing Perspective for Quantum Machine Learning Classification Advantage in Finance Using NISQ Algorithms"

_entropy, 2022, doi:10.3390/e24111656_

Round 1

Reviewer 1 Report

 Quantum machine learning is an interesting and important topic in quantum computing. The authors shows that they can achieve better classical encoding and performance of quantum classifiers by using Linear Discriminant Analysis (LDA) during the data preprocessing step.

This work is interesting and meaningful. It is well written and very comfortable to read.  Therefore, I recommend the author to revise this paper.

There are some issues that need to be fixed. 

The title of this paper is A preprocessing perspective for quantum machine learning classification advantage using NISQ algorithms.  However, the authors only use two datasets and they are related to bank system.  It is only enough to support the result claimed in the title.  

The author takes a lot contents to point out the datasets in page 2.  The datasets collected in Taiwan from 60 April 2005 to September 2005.  It is a pretty short time datasets and it comes from an island of China. It is hard to support the result  classification advantage using NISQ algorithms.   

Figure 1, page 4:  The figure looks beautiful.  However, the figure is to small such that the information in the figure is hard to read, it is hard to read even I zoom in.  Please make appropriate modification.  

Line 150, page 4:  belta_n and   x_n.    Should them be  belta_i and   x_i ? What are the other coefficients of belta_1 to  belta_{n-1} and x_1 to x_{n_1}

Line 153, page 5: P are the probability.     P is the probability or P are the probabilities.  

Line 158,  page 5:  a decision tree (CART).   What does CART means? Is it an abbreviation for decision tree?  But why it is C-A-R-T?  Why not DT.    

One Line before line 159, page 5:  put a comma at the end of the equation.  Similar revision in others places should be done in this paper.

Line 162,  page 5: Put a comma at the end of the equation and change Where  to where. Similar revision in others places should be done in this paper.

Line 163, page 5: put a a comma at the end of the line.    .. A and B,  respectively.

Author Response

Dear reviewer,

Thanks for the feedback and to provide excellent comments that will impact on the improvement of the article. Regarding the changes that we made based on your suggestions:

1. "The author takes a lot contents to point out the datasets in page 2.  The datasets collected in Taiwan from 60 April 2005 to September 2005.  It is a pretty short time datasets and it comes from an island of China. It is hard to support the result  “classification advantage using NISQ algorithms". CHANGE: The title was modified to include "in finance" to be more precise.
2. "Figure 1, page 4:  The figure looks beautiful.  However, the figure is to small such that the information in the figure is hard to read, it is hard to read even I zoom in.  Please make appropriate modification." CHANGE: We increased the size of the font in 2 to 4 points in all the texts included in the figure.
3. "Line 150, page 4:  “belta_n” and  “ x_n”.    Should them be  “belta_i” and  “ x_i” ? What are the other coefficients of belta1 to  belta{n-1} and x1 to x{n_1}". ANSWER: Beta are coefficients from 0 to n, and i is the sample, so we fit a linear regression with n features, so n beta for all the i sample.
4. "Line 153, page 5: P are the probability.     P is the probability or P are the probabilities." ANSWER: P is the probability.
5. "Line 158,  page 5:  a decision tree (CART).   What does CART means? Is it an abbreviation for decision tree?  But why it is C-A-R-T?  Why not DT." ANSWER: CART stands for Classification and Regression Trees (the modern name of Decision Tree).
6. "One Line before line 159, page 5:  put a comma at the end of the equation.  Similar revision in others places should be done in this paper." CHANGE: Corrected.
7. "Line 162,  page 5: Put a comma at the end of the equation and change “Where ” to “where”. Similar revision in others places should be done in this paper." CHANGE: Corrected.
8. "Line 163, page 5: put a a comma at the end of the line.    “.. A and B,  respectively”." CHANGE: Corrected.

Thanks again.

Kind regards,

Javier Mancilla

Reviewer 2 Report

The paper uses several preprocessing processes to reduce the dimension of the data structure. Especially they use the LDA and compare it with the PCA method. Finally, they demonstrate their approach to quantum machine learning classification in some real datasets. The paper is well-written and scientifically sound. Here, I suggest some modifications to improve the paper:

(1) The metrics presented in Table 1 should be clearly clarified the meaning.  If so, the readers can easy to follow the results in Figs. 2-5 and tables.

(2)The authors should clarify detailed how to compress over 100 features into two features (two qubits). Also, a comparison with a higher number of qubits would strengthen the paper. 

(3) Minor English should be revised, such as "data" is the plural, not singular.

Author Response

Dear reviewer,

Thanks for the feedback and to provide excellent comments that will impact on the improvement of the article. Regarding the changes that we made based on your suggestions:

1. "(1) The metrics presented in Table 1 should be clearly clarified the meaning. If so, the readers can easy to follow the results in Figs. 2-5 and tables." CHANGE: We wrote an specific paragraph before table 1 explaining the metrics.
2. "(2)The authors should clarify detailed how to compress over 100 features into two features (two qubits). Also, a comparison with a higher number of qubits would strengthen the paper. CHANGE: This is actually a great suggestion, and we considered that idea previously, but since testing with more qubits is very time consuming it's impossible to include this in the article now. Our purpose overall is to ignite the QML supervised learning efforts, so we hope to keep expanding our research on this (including more qubits and dealing with the barren plateau issues). 
3. "(3) Minor English should be revised, such as "data" is the plural, not singular.". CHANGE: The text was revisited and reviewed to correct the typos an grammatical errors.

Thanks again.

Kind regards,

Javier Mancilla

Reviewer 3 Report

This article is very well written. I only have a few minor comments.

Grammar: consider rewriting the sentence in line 131: “In the case of Pennylane’s default qubit, is a simple state-vector qubit simulator 131 designed in Python with JAX, Autograd, Torch, and Tensorflow.” Consideration rewriting the sentence in line 151 “Linear regression fails in the classification task is why the linear regression formulation is embedded in a logistic function such as Eq. 1 to compute a probability.”

Author Response

Dear Reviewer,

Thanks for the comments. Here the adjustments:

"Consider rewriting the sentence in line 131: “In the case of Pennylane’s default qubit, is a simple state-vector qubit simulator 131 designed in Python with JAX, Autograd, Torch, and Tensorflow.” 

>> Sentence rewritten 

Consideration rewriting the sentence in line 151 “Linear regression fails in the classification task is why the linear regression formulation is embedded in a logistic function such as Eq. 1 to compute a probability.”

>> Sentence rewritten and replaced by “Linear regression is used in regression tasks, but the method can be applied in the classification task by a subtle embedding in a logistic function such as Eq.1 to compute a probability”

Kind regards,

Javier Mancilla